# Thinning Effect of Few-Layer Black Phosphorus Exposed to Dry Oxidation

**DOI:** 10.3390/nano15130974

**Published:** 2025-06-23

**Authors:** Qianyi Li, Hang Yang, Xiaofang Zheng, Yu Chen, Chuanxin Wang, Yujie Han, Yujing Guo, Xiaoming Zheng, Yuehua Wei

**Affiliations:** 1Hunan Provincial Key Laboratory of Intelligent Sensors and Advanced Sensor Materials, School of Physics and Electronic Science, Hunan University of Science and Technology, Xiangtan 411201, China; 2208040428@mail.hnust.edu.cn (Q.L.); chen_yu@mail.hnust.edu.cn (Y.C.); 2408040305@mail.hnust.edu.cn (C.W.); 2School of Physics and Optoelectronics, Xiangtan University, Xiangtan 411105, China; 3Beijing Blue Sky Innovation for Frontier Science, Beijing 100085, China; yanghang10@nudt.edu.cn; 4Institute of Environmental Science, Shanxi University, Taiyuan 030006, China; 202212901005@email.sxu.edu.cn (X.Z.); yujiehan@sxu.edu.cn (Y.H.); guoyj@sxu.edu.cn (Y.G.)

**Keywords:** BP, dry oxidization, layer-by-layer thinning, p-type doping

## Abstract

Few-layer black phosphorus (BP) holds significant potential for next-generation electronics due to its tunable bandgap and high carrier mobility. The layer modulation of BP is essential in the applications of electronic devices ascribed to its thickness-dependent electronic properties. However, precisely controlling its thickness still presents a challenge for optimizing performance. In this study, we demonstrate that BP can be precisely thinned when exposed to dry oxygen (40% humidity, low oxygen concentration) in a dark environment, which is different from that exposed to humid oxygen (100% humidity, low oxygen concentration) without light illumination. The thinned BP not only demonstrates enhanced stability but also exhibits significant improvements in its electrical properties. The variation in bandgap from 0.3 to 2 eV, resulting in the *I*_ON_/*I*_OFF_ ratio increased from 10^3^ to 10^6^, and the hole mobility improved from 235 cm^2^ V^−1^ s^−1^ to 851 cm^2^ V^−1^ s^−1^, was ascribed to the layer-by-layer thinning and p-type doping effects induced by the formed P_x_O_y_. Our finding demonstrates significant potential of BP in future nanoelectronic and optoelectronic applications.

## 1. Introduction

Atomically thin black phosphorus (BP) has demonstrated significant potential for applications in field-effect transistors (FETs) [1,2,3], optoelectronics [4,5], photocatalysis [6], and biosensing applications [7,8]. Especially for transistor logic circuit, BP is preferable to its previous counterparts (graphene [9] and transition metal dichalcogenides (TMDs [10])), driven by the large carrier mobilities and high *I*_ON_/*I*_OFF_ ratio [11,12]. Unlike gapless graphene, BP possesses a relatively large adjustable bandgap, ranging from 0.3 eV to 2 eV, which ensures a high *I_ON_/I_OFF_* ratio [13]. Furthermore, BP is predicted to have a significantly lighter electron effective mass compared to TMDs, resulting in superior electron mobility. These properties promote BP as an excellent material for semiconducting devices. Recent studies, including Zhong’s summary, highlight BP oxidation complexity [14], while Lu et al. proposed a two-step method combining ultraviolet ozone oxidation and argon plasma etching for controlled thinning and enhanced FET performance [15].

Although BP exhibits superior properties over some 2D materials, the difficulty in precisely controlling its thickness and its ambient instability largely impedes its widespread application [16,17]. For example, the stable monolayer BP cannot be easily obtained via mechanical exfoliation, and the thickness of BP flakes cannot be controlled by this method [18], due to its strong interlayer coupling and fragile nature. The performance of BP devices is degraded rapidly when exposed to the surrounding environment [17,19]. Experiments have shown that exposure to ambient conditions leads to the formation of a rough surface on BP, containing impurities that serve as charge trapping and carrier scattering centers. As a result, the hole and electron mobilities decrease by more than two orders of magnitude and the *I*_ON_/*I*_OFF_ ratio of BP drops from 10^3^ to below 10 upon exposure to air [20]. This degradation mechanism can be generally attributed to its easy reaction with water and oxygen under light illumination in air. Ziletti et al. proposed an oxidation pathway for BP involving a triplet-to-singlet conversion of oxygen molecules, which lowers the oxygen dissociation barrier and significantly increases the likelihood of oxidation [21]. Under typical conditions, incident light serves as the excitation source to initiate the chemical reaction between BP and oxygen. More recently, Hu et al. proposed that BP could be oxidized by water vapor and oxygen, even in a dark environment [17]. The formation of oxidation products, such as P_x_O_y_, emerge as pits and bubbles on the BP surface, which significantly increase the surface roughness, ultimately leading to the degradation of BP FETs performance. In fact, if the oxidation rate can be controlled and reduced, it may be possible to minimize the formation of pits and bubbles, leading to improved electrical performance of BP.

In this study, we investigate the evolution of electrical properties of few-layer BP exposed to dry oxygen (40% humidity, low oxygen concentration) under dark conditions. BP demonstrates relatively better stability compared to humid oxygen (100% humidity, low oxygen concentration) and can be thinned layer-by-layer, resulting in the variation in bandgap from 0.3 to 2 eV. In addition, the electrical performance of BP FETs can be significantly enhanced. The *I*_ON_/*I*_OFF_ ratio increased from 10^3^ to 10^6^ during the thinning process, and the hole mobility improved from 235 cm^2^ V^−1^ s^−1^ to 851 cm^2^ V^−1^ s^−1^, attributed to the charge transfer from BP to P_x_O_y_, as confirmed by photoemission spectroscopy. Our research supports the fabrication of few-layer BP FETs with improved performance.

## 2. Results and Discussion

BP is a single-elemental, layered crystalline material consisting solely of phosphorus atoms, exhibiting a puckered honeycomb structure, with interlayers bonded by van der Waals forces [22]. The crystalline structure is shown in Figure 1a. To achieve high mobility and a high switching ratio (on/off current ratio), BP with a thickness of 5–15 nm is typically selected as the device channel. Figure 1b shows the atomic force microscopy (AFM) image. As the height profile curve demonstrated, the thickness is about 12 nm (~20 layers). Three predominant Raman peaks ($A_{g}^{1}$, $B_{g}^{2}$ and $A_{g}^{2}$) located at ~360, ~437 and ~466 cm^−1^ (Figure 1b) are observed, which is in good agreement with previous literature [4]. The transfer characteristics of pristine BP FET (the red line) are shown in Figure 1c, where the current initially decreases with the gate voltage ranging from −80 V to −20 V, and then increases as the gate voltage is swept from −20 V to +50 V. Furthermore, the drain current at +50 V is of the same order of magnitude as that at −50 V, suggesting an ambipolar transistor characteristic. Consequently, BP FETs can operate in both hole and electron doping regimes by controlling the gate voltage. In addition, the current level can be modulated over three orders of magnitude, with OFF currents on the order of tens of nA and ON currents on the order of tens of µA. Notably, a large hysteresis window (~20 V, Appendix A) is observed between the forward and backward sweep curves, indicating the presence of interface trap effects from SiO_2_. Additionally, the output curves stay linear (Appendix A), suggesting the good ohmic contact between BP and metal electrodes. These intrinsic features of BP are consistent with results reported in the previous literature.

To investigate the effects of humid oxygen on the transport properties of BP FETs, we first examined the evolution of the electrical properties of the BP FET exposed to humid oxygen (100% humidity, low oxygen concentration, at room temperature ~25 °C) in a dark environment. All electrical measurements were conducted under high vacuum (10^−6^ mbar) and in dark conditions. The transport behavior of BP FET was monitored and demonstrated in Figure 1c. Consequently, the on-state current (hole conduction region) of the BP FET drops sharply from 22 μA to less than 0.1 μA after 24 h of exposure. The hole mobility and *I*_ON_/*I*_OFF_ ratio of the BP FET as a function of exposure time can be extracted (Figure 1d). It was observed that the *I*_ON_/*I*_OFF_ ratio drops from 10^3^ to less than 10, and the hole mobility decreases from 400 cm^2^ V^−1^ s^−1^ to 2 cm^2^ V^−1^ s^−1^ when exposed for 24 h, indicating the severely declining performance of BP FET exposed to humid oxygen. This degraded electrical performance in BP can be attributed to its reaction with water and oxygen in a dark environment. This is because the energy levels of O_2_ are lowered when water molecules approach oxygen, which facilitates electron transfer from BP to oxygen and triggers BP oxidation even in dark conditions [17].

Considering the degradation in the performance of BP exposed to humid oxygen in a dark environment, we then investigated the effect of reducing humidity to 40% (dry oxygen) on the electrical properties of BP under dark conditions. Figure 2a illustrates the evolution of transfer characteristic curves for ~12 nm thick BP FET as a function of exposure time to dry oxygen. It is evident that the open-state current of the transfer characteristic curves does not undergo significant degradation after being exposed to dry oxygen for 60 h, demonstrating good stability. The current gradually decreases with increasing exposure time (Appendix A). Additionally, the minimum current of BP FET dramatically varies with increasing oxidation time, shifting towards negative gate voltages from −20 V (pristine) to −60 V (after 84 h of exposure to dry oxygen). To further investigate the influence of dry oxygen exposure time on BP electrical properties, the key performance parameters were extracted. As shown in Figure 2b, the hole mobility increases sharply from ~235 cm^2^ V^−1^ s^−1^ to ~851 cm^2^ V^−1^ s^−1^ during the first 36 h. Then it gradually decreases to less than ~50 cm^2^ V^−1^ s^−1^. Meanwhile, the sub-threshold swing (SS) shows a decreasing trend, ranging from ~10.1 V/dec to ~1.8 V/dec (the left axis in Figure 2c), indicating that the Fermi surface of BP can be effectively regulated by gate voltage [23,24]. Accordingly, it can be concluded that the electrical performance of BP can be improved by exposure to dry oxygen, highlighting the significant discrepancy between ambient exposures for BP FET [16,17,24].

It is worth noting that the open-state current of the transfer characteristic curve underwent insignificant degradation, decreasing from 10^−5^ A to 10^−6^ A, while the off-state current decreased from 10^−8^ A to 10^−12^ A (Figure 2d), resulting in the increase of *I*_ON_/*I*_OFF_ ratio from 10^3^ to 10^5^. This apparent change in the switching ratio implies a variation in the bandgap of BP during oxidation. Here, the bandgap can be extracted according to the threshold voltages of the ambipolar transfer curves [25]:$$E_{g}=e\left(V_{DS}+\frac{V_{n-th}-V_{p-th}}{\beta }\right)$$
where e is the elementary charge, *V_n-th_* and *V_p-th_* are the threshold voltage of n-branch and p-branch, respectively. The detailed analysis method of the calculated *E_g_* are displayed in Appendix A. As demonstrated in Figure 2c, the estimated bandgap of BP increases from 0.3 eV to 2 eV as a function of dry exposure time, indicating an apparent enlargement of the bandgap in BP exposed to dry oxidation. As we know, the bandgap is associated with thickness. Such change in bandgap reflects the various thickness of few-layer BP exposed to dry oxidation process. In addition, the nonmonotonic trend of the on-state current for thinned BP can be easily explained using a resistive network model (inserted in Figure 2d), which incorporates Thomas–Fermi charge screening and interlayer resistive coupling, and has been successfully used to describe the tunable transport properties of BP with varying thicknesses [13,26]. Thus, these results suggest that the thinning effect may occur in BP exposed to dry oxidation.

To further investigate the thinning effect of BP during dry oxygen exposure, the electrical properties of intrinsic BP FETs with different thicknesses (ranging from 1 nm to 20 nm) were investigated. The corresponding transfer characteristics are presented in Figure 2e. Interestingly, the *I*_OFF_ current significantly varies from 2.3 × 10^−12^ to 6.7 × 10^−8^ with the increasing thickness while only minor variation occurs in *I*_ON_ current (from 2.8 × 10^−7^ to 1.5 × 10^−5^), thus leading to bandgap-related enhancement of *I*_ON_/*I*_OFF_ ratio from 10^3^ to 10^5^. It is worth noting that the *I*_ON_/*I*_OFF_ ratio of pristine BP with respect to thickness displays the same trend as that of few-layer BP exposed to dry oxidation (Figure 2b). The extracted electrical properties for pristine BP FETs are shown in Figure 2f. The mobility firstly increases from 148 cm^2^ V^−1^ s^−1^ (~20 nm) to 385 cm^2^ V^−1^ s^−1^ (~8 nm), and then decreases to less than 10 cm^2^ V^−1^ s^−1^ (~1 nm). The thickness dependence of mobility can be attributed to that the thinner channel on the substrate is more susceptible to potential variations due to the formation of traps at the interface, which has been observed in other layered 2D materials [27,28]. The mobility increases monotonically with the thickness of BP ranging from 1 nm to 8 nm, attributed to the enhanced charge screening. However, the mobility continuously decreases when the thickness increases from ~10 nm to ~20 nm, derived from the rise in interlayer resistance for BP [13]. The trend of change in intrinsic BP as a function of layers is consistent with that of few-layer BP exposed to dry oxygen, further identifying that the thinning effect of BP occurs during exposure to dry oxygen.

Raman spectra are highly sensitive to the variation in layer number of two-dimensional materials, which makes them an effective tool for identifying the thinning effect of BP exposed to dry oxygen. Figure 3a depicts the evolution of Raman characteristics of ~20 nm thick BP exposed to dry oxidation. The relative Raman intensity of BP versus Si gradually decreases as the exposure time increases to 12 h. Furthermore, the Raman peaks of BP ($A_{g}^{1}$, $B_{g}^{2}$ and $A_{g}^{2}$) significantly decrease as the exposure time is extended, eventually nearly disappearing after 84 h of exposure to dry oxygen. The quenching of the Raman intensity identifies the layer variation in BP exposed in dry oxygen [28,29]. Additionally, considering the $A_{g}^{2}$ peak position of few-layer BP are sensitive to the layer number [27], thus the $A_{g}^{2}$ peak Raman intensity are normalized to further study the thinning effect of BP exposed in the dry oxygen, as shown in Figure 3b. The $A_{g}^{2}$ Raman peaks show a noticeable redshift from ~476 cm^−1^ to ~461 cm^−1^ (Appendix A), confirming that the BP is thinned. This phenomenon is consistent with previous studies [16,29,30,31].

Considering that BP is exposed to dry oxygen with ultra-low water concentration, relatively low oxygen concentration, and no light illumination, we propose a two-step degradation mechanism for the dry oxidation process of BP, which involves: (i) The generation of superoxide in the first layer upon oxygen chemisorption. (ii) The dissociation of superoxide in the presence of water to form an O_2−_·H_2_O cluster, followed by the complete oxidation of the top layer to form P_x_O_y_. In other words, the surface of BP is firstly oxidized by oxygen, and then the water molecules react with the superoxide to form fully oxidized products (P_x_O_y_). As O_2_ further diffuses into the P_x_O_y_ layer and oxidizes the underlying phosphorene, the BP flakes will eventually be fully oxidized layer by layer. The schematic of the layer-by-layer thinning mechanism is shown in Figure 3c. Initially, the oxides (P_x_O_y_) form on the topmost layer of the BP surface. As the exposure time increases, the oxidation process progresses, resulting in layer-by-layer oxidation of the subsequent BP layers. This process of dry oxidation is different from that in the ambient environment. In the latter, a high concentration of water vapor plays a dominant role, leading to the rapid deformation and disintegration of the structure (typically observed as a bumpy surface of the resulting oxides), a process referred to as “well-oxidation” rather than layer-by-layer oxidation [32]. The dry oxidation process also differs from oxidation under pure oxygen [33], as a small amount of water slowly destroys the top layer of BP, enabling layer-by-layer oxidation, which is consistent with the results of DFT calculations [19]. In addition, compared to traditional methods [18,34,35], our oxidation approach enables more effective layer-by-layer thinning while protecting the surface oxide layer, offering a more controlled and energy-efficient solution for few-layer BP processing.

To further investigate the doping process and mechanism, X-ray photoemission spectroscopy (XPS) characterization was performed on bulk BP exposed to dry oxygen. Considering the large light spot in millimeter scale, the bulk BP was substitute for the mechanically exfoliated few-layer BP. Figure 4a shows P 2p and O 1s core level spectra of bulk BP as a function of the exposure time. All spectra were calibrated to the binding energy of carbon (284.8 eV). Two broad peaks located at 129.6 and 130.4 eV can be observed in P 2p core level spectra for pristine BP, corresponding to P 2p 3/2 and P 2p 1/2, respectively. As the exposure time increased to 12 h, the intensity of the P 2p peak gradually decreased, and a new peak at 135.0 eV appeared, indicating the formation of P_x_O_y_ [36,37]. The intensity of P_x_O_y_ peak consistently increases with increasing exposure time. Furthermore, the intensity of the O-associated peaks at 533.6 eV also continues to increase. These results confirm the oxidation of BP during the dry oxidation process. In the P 2p 3/2 core level spectrum of bulk BP (Figure 4b), a shift to lower binding energy of approximately 0.2 eV is observed after 84 h of exposure, indicating that the Fermi level of BP moves towards or even above its valence band maximum (VBM). This phenomenon certifies the p-type doping of BP exposed to dry oxygen. The doping mechanism was further investigated using the ultraviolet photoemission spectroscopy (UPS) of bulk BP as a function of dry oxidation time. As shown in Figure 4c, the vacuum level of BP was measured to extract the work function by linearly extrapolating the low kinetic energy onset (secondary electron cutoff). Compared with pristine BP, the work function increased by approximately 0.1 eV after exposure for 24 h. When exposed for 48 h, the work function increases to 5.1 eV. The significant difference in work function between P_x_O_y_ and BP leads to a considerable charge transfer from BP (with a lower work function) to P_x_O_y_ (with a higher work function), resulting in the accumulation of excess delocalized holes in the BP layer. This is consistent with the XPS results shown in Figure 4a. Therefore, the thinning and doping effects contribute to the enhanced electrical properties of BP exposed to dry oxidation, with the hole mobility of the BP device exposed to dry oxidation being higher than that of the BP device exposed to pure oxygen. In addition, it should be noted that the existence of P_x_O_y_ in BP may introduce some trap states, leading to the left-shift in the neutral point in the electrical characterization driven by the increase in hysteresis (Appendix A). Notably, such large hysteresis is expected to open new avenues for important applications in memristors.

## 3. Conclusions

In conclusion, we observed that the oxidation process of few-layer BP in dry oxygen was distinctly different. After placing the few-layer BP in dry oxygen, it not only exhibited long-term stability but also showed improvements in the *I*_ON_/*I*_OFF_ ratio, hole mobility, and sub-threshold swing. Through thickness-dependent experiments, Raman characterization, and in situ XPS/UPS measurements, we propose two mechanisms responsible for this interesting phenomenon: the layer-by-layer thinning effect and p-type doping effect induced by the formation of P_x_O_y_. This new degradation mechanism lays an important foundation for the development of proper protecting schemes in black phosphorus-based devices.

## 4. Experimental Section

Sample preparation and device fabrication: Scotch tape was used to mechanically exfoliate BP flakes from bulk BP crystals (Shanghai Onway Technology Co., Ltd., Shanghai, China). The exfoliated BP flakes were then rapidly transferred onto a degenerately p-type doped silicon substrate with 285 nm SiO_2_. After mechanical exfoliation, the target BP flakes on the substrate were located using an optical microscope, and polymethylmethacrylate (PMMA) photoresist was subsequently spin-coated onto the silicon substrate for further processing. Subsequently, the BP FET was fabricated by standard micro-nano fabrication techniques. The source and drain contacts were defined by e-beam lithography (EBL, Eline Plu Raith company, Dortmund, Germany) and formed by depositing a 5 nm Ti/50 nm Au metal stack using thermal evaporation (PVD75, Kurt J. Lesker Company, Jefferson Hills, PA, USA). Finally, the fabricated devices were wire-bonded onto a leaded chip carrier for subsequent electrical measurements. All BP FETs were fabricated and pre-annealed in a nitrogen glovebox. Prior to environmental exposure tests, the pristine devices were characterized in high vacuum (10^−6^ mbar) to establish baseline electrical properties. The initial humidity/oxygen history of all samples was identical.

Characterization and electrical measurement:

The mobility *μ* and the sub-threshold swing *SS* are obtained from:$$\mu =\frac{dI_{DS}}{dV_{G}}\frac{L}{WV_{DS}C_{OX}}$$$$SS=\frac{d\log\left(I_{DS}\right)}{dV_{G}}$$
where *I*_DS_ is the drain-source current, *V_G_* is the gate voltage, *V_DS_* is the drain-source voltage, *L* and *W* represent the channel length and width of device, and *C_OX_* is the gate capacitance.

The topography of the samples was characterized via atomic force microscopy (AFM, Bruker company, Santa Barbara, CA, USA, scanning mode: semi-contact, scanning frequency: 1.01 Hz) and high-resolution optical microscopy (Nikon Eclipse LV100D, New York, NY, USA). The Raman spectra were recorded using a Confocal Raman Spectrometer (WiTec company, Ulm, Germany, spot size: 2 µm). The excitation source was a 532 nm laser (2.33 eV) with a power below 0.1 mW to avoid laser-induced heating and damage. All characterizations were conducted in ambient conditions at room temperature (300 K).

UPS and XPS measurements of bulk BP were carried out in a customer-designed ultrahigh vacuum system (In situ UPS/XPS investigations were carried out in a home-built ultrahigh vacuum system (10^−10^ mbar) at room temperature. The excitation source for UPS and XPS measurements were He Iα (*hν* = 21.2 eV) and Al Kα (*hν* = 1486.7 eV), respectively. The position of the vacuum level was determined from the low kinetic energy region of the UPS spectra, with a sample bias of −5 V. The sample work function *φ* was calculated using the equation *φ = hν − W*, where *W* is the spectrum width (the energy difference between Fermi level and second electron cut off). The φ of the electron analyzer was measured to be 4.30 ± 0.05 eV. The air exposure of the sample was conducted in the load lock chamber under dark environment. After air exposure, the load lock chamber was pumped down to ultrahigh vacuum and the sample was transferred to analyzer chamber for characterization.

The devices were loaded in a custom-designed high-vacuum system (base pressure ~10^−6^ mbar) for subsequent electrical measurements. The corresponding electrical measurements were conducted by an Agilent 2912A source measure unit at room temperature. For the air-oxidation process, the BP device was first exposed to dry oxygen (25 °C and 40% humidity) in the dark and then evacuated to a high vacuum for electrical measurement.

## Figures and Tables

**Figure 1 nanomaterials-15-00974-f001:**
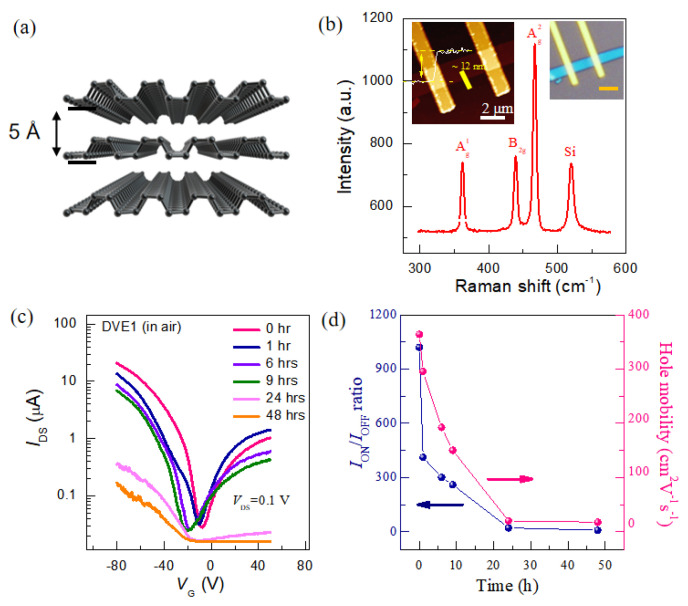
Typical characterizations of a 12 nm BP sample exposed to humid oxygen without light illumination. (**a**) Crystal lattice structure of few-layer BP. (**b**) Raman spectrums of the few-layer BP flake. The inserts show the corresponding device image and the AFM. (**c**) Evolution of transfer characteristics of BP FET as a function of exposure time. (**d**) Extracted hole mobilities and *I*_ON_/*I*_OFF_ ratios versus exposure time.

**Figure 2 nanomaterials-15-00974-f002:**
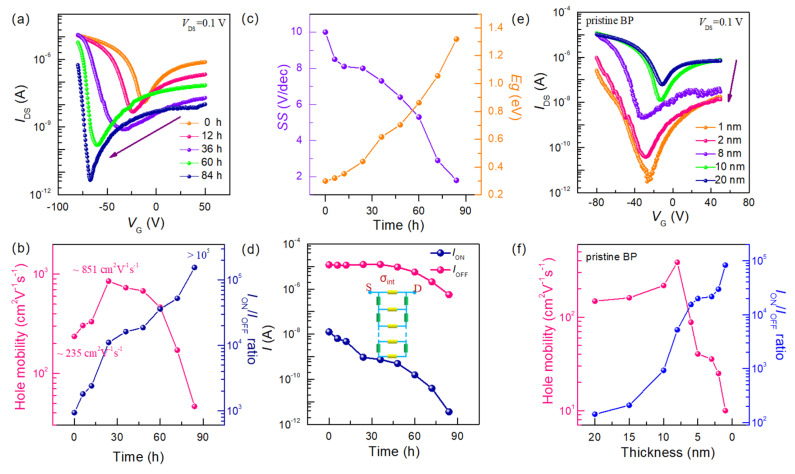
Electrical properties of 12 nm BP FET exposed to dry oxygen (0 s to 84 h) and pristine BP with varying thicknesses. (**a**) Transport characteristics dependent on oxidation time, with V_d_ = 0.1 V, presented on a logarithmic scale. (**b**) Extracted key parameters as a function of oxidation time: the hole mobility (left axis) and *I*_ON_/*I*_OFF_ ratio (right axis). (**c**) Sub-threshold swing (SS) and calculated bandgap. (**d**) *I*_ON_ current and *I*_OFF_ current. The inset shows the resistive network model, which includes Thomas–Fermi charge screening and interlayer coupling, that describes the nonmonotonic trend of the ON current. (**e**) Transfer characteristics of pristine BP FET with varying thicknesses. (**f**) *I_ON_/I_OFF_* ratio and hole mobility of pristine BP as a function of thickness.

**Figure 3 nanomaterials-15-00974-f003:**
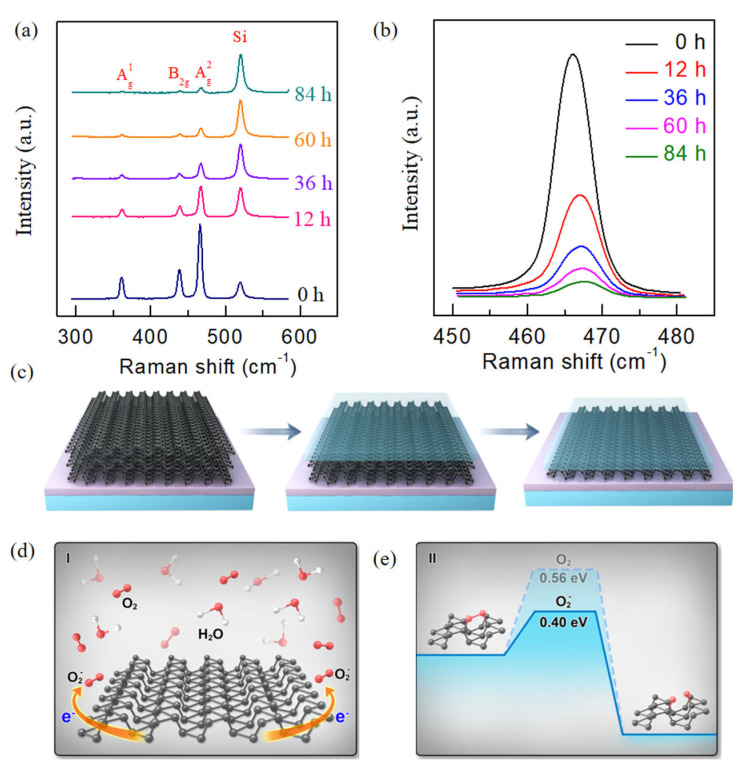
Mechanism of enhanced electrical property in BP FET exposed in dry oxygen. (**a**) Evolution of Raman spectra of few-layer BP in dry oxidation process. (**b**) A local magnification diagram of $A_{g}^{2}$ peaks. (**c**) Thinning schematic of BP exposed in dry oxidation. (**d**,**e**) Two-step reaction path for the oxidation process of the BP. The first step involves charge transfer process from BP to O_2_ in the presence of water to form O_2−_·H_2_O cluster, and then the active species of O_2−_.

**Figure 4 nanomaterials-15-00974-f004:**
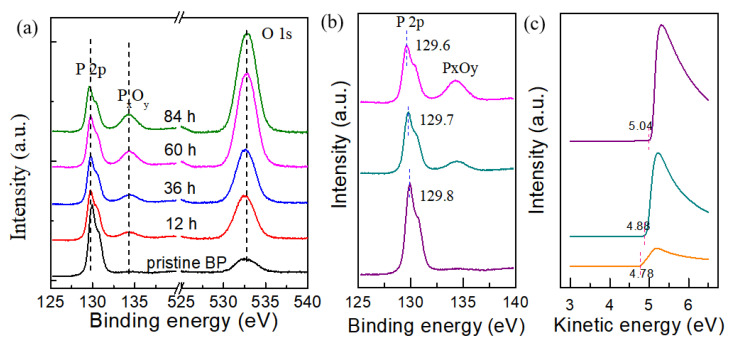
XPS and UPS characterizations of the bulk BP in dry oxidation process. (**a**) XPS spectra of P 2p and O 1s core level with respect to the exposure time. (**b**) P 2p core level of bulk BP. (**c**) The secondary electron cutoff regions of UPS spectra.

## Data Availability

Data are contained within the article and Appendix A.

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
