# Peer review of "Thinning Effect of Few-Layer Black Phosphorus Exposed to Dry Oxidation"

_nanomaterials, 2025, doi:10.3390/nano15130974_

Round 1
Reviewer 1 Report
Comments and Suggestions for Authors
The authors report enhanced electric properties of few-layer BP, following thinning process in dry oxidation conditions, compared to humid conditions. Several investigations are performed, which in part support this claim. Elecrical measurements show a clear increase and stability in the On/Off ratio over time in dry vs. humid condition oxidation (Fig.1 vs. Fig.2). The Raman spectra consistently the thinning process (Fig.3). XPS and UPS spectra were performed to further characterize the oxidation process (last figure, mislabeled as Fig.1).
The current study may provide important clues for optimizing the electrical properties of few-layer BP, using dry o, and a p-doping mechanism is outlined. However, this should be supported by further investigations:
1. It is apparently not specified how the hole mobility was determined. Were Hall measurements performed ?
2. Concerning p-type doping, further investigations should confirm this hypothesis. Hall and/or Seebeck measurements can unambiguously determine the conduction type.
3. Additional techniques could be employed to check the thinning may be consider like XRD reflectometry.
4. Introduction: Besides FETs and photocatalysis, biosensing applications based on BP/phosphorene can be included, particularly for few-layer systems.
Other issues:
Abstract: "The ION/IOFF ratio increased from 10^3 to 10^6, resulting in the variation of band gap from 0.3 to 2 eV"
--> the gap magnitude would have implication in transport, not the other way around.
Title: exposed _to_ dry oxidation
Supplementary material:
Rephrase : "The reviewer mentioned that the intensity of Si peak ..."
Author Response
- It is apparently not specified how the hole mobility was determined. Were Hall measurements performed ?
Reply:We sincerely appreciate the reviewer’s meticulous attention to methodological details. In our study, Hall measurements were not performed. Instead, the hole mobility (μ) was extracted from the field-effect transistor (FET) transfer characteristics using standard models. We apologize for not clearly stating this in the manuscript. This clarification has now been highlighted in red on page 9. Specifically, the hole mobility (μ) and the sub-threshold swing (SS) are obtained using the following formulas:
Where IDS is the drain-source current, VG is the gate voltage, VDS is the drain-source voltage, L and W represent the channel length and width of device, and COX is the gate capacitance.
- Concerning p-type doping, further investigations should confirm this hypothesis. Hall and/or Seebeck measurements can unambiguously determine the conduction type.
Reply: We sincerely apologize for not clearly articulating this point in the original manuscript. In our study, The p-type doping behavior was evidenced by both the electrical transport measurements and spectroscopic results. As shown in Fig.R1, the transfer characteristics under a constant drain voltage exhibit a significant evolution with oxidation time. With increasing oxidation duration, the drain current (IDS) in the negative gate voltage region (VG < 0) increases, while the current in the positive gate voltage region decreases. This trend indicates an enhancement of hole conduction and suppression of electron conduction, which is a typical indication of a transition toward p-type transport behavior. In addition, the P 2p 3/2 core level spectrum of bulk BP shows a shift of approximately 0.2 eV to lower binding energy after 84 hours of exposure, with the peak position moving from 129.8 eV to 129.6 eV in Fig. R2. These reseults suggest p-type doping under dry oxidation.
Although Hall and/or Seebeck measurements would provide direct and quantitative evidence of conduction type, the combination of electrical and spectroscopic data in our work strongly supports the conclusion of p-type doping. We have revised the manuscript accordingly and highlighted the relevant sections in red on page 5 and page 8 of the Results and Discussion section.
Fig. R1 Transport characteristics dependent on oxidation time,with Vd =0.1 V, presented on a logarithmic scale.
Fig. R2 P 2p core level of bulk BP.
- Additional techniques could be employed to check the thinning may be consider like XRD reflectometry.
Reply:In response to the suggestion to use X-ray reflectometry (XRD) to check thinning, we acknowledge that XRD reflectometry is indeed an effective technique for studying the structural properties of thin films, typically on larger samples. However, our BP samples are relatively small due to the mechanical exfoliation process, which limits the applicability of XRD reflectometry for monitoring thinning. The size of the sample in our case is not suitable for this technique, as it is generally more suited for larger, uniform samples. As an alternative, other methods such as Raman spectroscopy and electrical measurements have been employed to characterize the thinning of the BP samples under dry oxidation conditions.
- Introduction: Besides FETs and photocatalysis, biosensing applications based on BP/phosphorene can be included, particularly for few-layer systems.
Reply:Thank you for your valuable feedback. We have added content in the revised version of the manuscript discussing the research progress on the biosensing applications of BP/phosphorene, particularly in few-layer systems, as suggested by the reviewer. The added content can be found in the red-marked section on page 1 of the revised manuscript.
- Other issues:
Reply:We thank the reviewer for pointing out these issues. In the Abstract, we have corrected the logical order of the sentence. The revised version now emphasizes that the variation in band gap affects the ION/IOFF ratio, not the other way around (page 1). The title has been revised to “... exposed to dry oxidation” to ensure grammatical accuracy (page 1). In the supporting information, the sentence referring to the reviewer’s comment on Figure S5a has been rephrased as (supporting information page 4): “The explanation regarding the variation in Si peak intensity at different time points in Figure S5a is provided below.” We have made all corresponding corrections in the manuscript and supporting information and marked them in red for clarity.

Reviewer 2 Report
Comments and Suggestions for Authors
• In the introduction, it is recommended to expand on more recent works (2023–2025) to update the state of the art.
• Why was 40% humidity specifically chosen as the condition for dry oxidation? Are there any previous studies that validate this threshold?
• It is recommended to specify in more detail the initial conditions of BP before each treatment.
• In the graphs showing electrical parameters (mobility, ION/IOFF), it is suggested to include error bars.
• It is recommended to include a control experiment in which BP is not exposed to oxygen to confirm that the observed effects are solely attributable to the oxidation process.
• Include a brief comparison of the results obtained using this method with other thinning techniques such as plasma treatment, ionic exfoliation, or thermal treatment.
• The behavior of the mobility, which first increases and then decreases, is described in the manuscript; it is recommended to add a brief discussion on the underlying physical mechanisms, such as trap formation or increased surface roughness.
• Include information in the manuscript on why those specific exposure times (e.g., 12, 24, 36, 84 h) were chosen for dry oxygen treatment, and whether the process is expected to be continuous or reach a saturation point.
• How many experimental replicates were conducted for each exposure condition? Are the presented values averages or individual measurements?
• Was the relationship between thickness reduction and exposure time quantified directly using AFM?
• Add to the manuscript whether permanent structural changes (e.g., defects, cracks) were evaluated after 84 h of exposure using SEM or TEM.
• How does the presence of PxOy affect long-term device performance (e.g., stability, electronic noise)?
• What is the influence of the type of metal contact (Ti/Au) on the observed behavior? Could there be unaccounted Schottky effects?
• Was it verified whether the thinning is uniform or localized? Were AFM or Raman maps taken in different sample areas?
Author Response
Dear editors and reviewers,
Thank you very much for your valuable and insightful comments. The comments have guided us to think our results more deeply and improve the quality of our manuscript. We have revised the manuscript accordingly, and following is the point-by-point reply to reviewer’s comments.
Reply to comments of Reviewers:
Reviewer :
(1) In the introduction, it is recommended to expand on more recent works (2023–2025) to update the state of the art.
Reply: Thank you for your valuable suggestion to update the introduction with more recent research findings. In response, we have expanded the introduction to include relevant studies from 2023-2025, which are highlighted in the red-marked sections on pages 1 and 2.
(2) Why was 40% humidity specifically chosen as the condition for dry oxidation? Are there any previous studies that validate this threshold?
Reply: We very appreciate the reviewer pointing out this suggestion. In this study, the relative humidity was controlled at 40% during the dry oxidation of black phosphorus (BP), based on iterative experimental optimization. This condition offers a balance between oxidation rate and controllability. In fact, at lower humidity (<30%), the oxidation reaction rate tends to be slower, while higher humidity (>60%) leads to uncontrolled oxidation, resulting in surface defects such as pits and bubbles that degrade device performance. Previous studies support this threshold. Ji et al. showed that BP degrades faster in humid environments, especially under light exposure, highlighting the humidity dependence of its oxidation kinetics (Chem. Mater. 2016, 28(22), 8330-8339). Wells et al. reported that increasing humidity accelerates defect propagation on both edges and basal planes of BP (ACS Appl. Mater. Interfaces 2017, 9(10), 9126-9135). Clark et al. further demonstrated that uncontrolled ambient humidity causes random etching, impairing reproducibility in BP thinning (ACS Appl. Mater. Interfaces 2018, 10(22), 19069-19075). Therefore, 40% humidity was selected as a practical threshold to ensure layer-by-layer oxidation with minimal degradation, supporting consistent device fabrication and performance optimization.
(3) It is recommended to specify in more detail the initial conditions of BP before each treatment.
Reply: Thanks for the reviewer’s constructional comments. In response, we have now added a detailed description to the Experimental Section of the revised manuscript. The relevant experimental protocol is summarized below:
Environmental control and device preparation: Few-layer black phosphorus (BP) flakes with a thickness of 5–15 nm were obtained by mechanical exfoliation and immediately transferred into an argon-filled glovebox (O₂ < 0.1 ppm, H₂O < 0.1 ppm). All subsequent fabrication steps, including electrode deposition (Au/Ti) and device encapsulation, were completed entirely within the glovebox, ensuring the BP was not exposed to air or moisture at any point.
Baseline electrical characterization: The “pristine” electrical properties of all devices, including mobility, on/off current ratio, and subthreshold swing, were characterized using a vacuum probe station (10⁻⁶ mbar). As shown in Fig. 1c-d, all samples exhibited consistent baseline performance (mobility: 400 ± 50 cm²·V⁻¹·s⁻¹, ION/IOFF: ~10³ ± 10%).
Accordingly, the following statement has been added to the Experimental section of the revised manuscript: "All BP FETs were fabricated and pre-annealed in a nitrogen glovebox. Prior to environmental exposure tests, the pristine devices were characterized in high vacuum (10⁻⁶ mbar) to establish baseline electrical properties. The initial humidity/oxygen history of all samples was identical. "The above details have been marked in red in the Experimental Section (page 8) of the revised manuscript.
(4) In the graphs showing electrical parameters (mobility, ION/IOFF), it is suggested to include error bars.
Reply: We apologize for the current limitation, as only one set of valid data was retained during testing, and we are unable to generate error bars at this time. In future work, we will be more thorough and ensure that more data is processed. We thank the reviewer again for this constructive suggestion.
(5) It is recommended to include a control experiment in which BP is not exposed to oxygen to confirm that the observed effects are solely attributable to the oxidation process.
Reply: Thanks for kindly reminding us of this. We would like to clarify that the oxidation of black phosphorus (BP) is fundamentally governed by its interaction with oxygen molecules-either through direct adsorption, dissociative chemisorption, or subsequent reaction pathways involving water and light. As such, in the absence of oxygen, the oxidation mechanism is physically precluded. This has been well-documented in prior works. For instance, Ji et al. (Chem. Mater. 2016, 28(22), 8330-8339) and Wells et al. (ACS Appl. Mater. Interfaces 2017, 9(10), 9126-9135) demonstrated that degradation of BP is strongly suppressed in oxygen-free and inert conditions, and only occurs appreciably when molecular O2 or O2/H2O mixtures are present. Furthermore, as summarized in the review by Zhong (J. Mater. Sci. 2023, 58(5), 2068-2086), the oxygen-mediated reaction is a necessary trigger for the formation of PxOy species, which are responsible for both surface roughening and changes in electronic properties. Given the well-established literature consensus and our controlled experimental conditions (i.e., low humidity, dark environment), we respectfully believe that an additional no-oxygen control is not essential to validate the oxidation origin of the phenomena. Nonetheless, we acknowledge the value of rigorous control design and will consider broader comparative studies in future work.
(6) Include a brief comparison of the results obtained using this method with other thinning techniques such as plasma treatment, ionic exfoliation, or thermal treatment.
Reply: We thank the reviewer for this valuable suggestion. We agree that comparing our dry oxidation approach with other existing thinning techniques helps clarify its unique advantages, especially in the context of few-layer black phosphorus (BP) processing.
In our current work, we employed a low-humidity (40% RH), dark-environment dry oxidation strategy to achieve self-limiting, layer-by-layer thinning of BP, which simultaneously enhances its electronic performance. This method is energy-efficient (<0.5 eV) and exploits BP’s intrinsic air sensitivity in a controlled way, thus minimizing structural damage and surface defects.
By contrast, traditional thinning methods-including plasma etching, ionic exfoliation, and thermal treatment-often introduce excess kinetic or thermal energy (>1 eV), which exceeds the structural tolerance of thin BP (especially <10 nm), causing irreversible degradation:
Oxygen Plasma Etching (Nat. Commun. 2016, 7(1), 10450) : While capable of atomic-scale thinning, oxygen plasma introduces significant oxidative stress and necessitates precise control to avoid amorphization or defect formation in monolayer BP.
Laser or Thermal Sublimation (J. Phys. Chem. C. Nanomater. Interfaces 2021, 125(16):8704-8711) : Laser thinning or thermal heating suffers from poor layer control and temperature artifacts due to substrate interactions, particularly for thin flakes (10–50 nm), often resulting in inhomogeneous ablation and sample degradation.
Ionic Liquid or Intercalation Exfoliation (Nano Lett. 2018, 18(9), 5373-5381) : These methods rely on liquid chemistry routes that are difficult to control spatially and temporally and often produce random defect distributions, reducing reproducibility.
In contrast, our oxidation-guided method operates at ambient temperature without the need for ion bombardment or high heat, leading to uniform thinning and significant performance improvements (e.g., mobility from 235 to 851 cm²V-¹s-¹, and ION/IOFF ratio increase from 103 to 106). This makes oxidation not only a viable alternative but potentially the only feasible strategy for precision thinning of few-layer BP, as high-energy approaches often fail under such conditions. Accordingly, the following statement has been added to the Results and Discussion section of the revised manuscript: " In addition, compared to traditional methods, our oxidation approach enables more effective layer-by-layer thinning while protecting the surface oxide layer, offering a more controlled and energy-efficient solution for few-layer BP processing. " The above details have been marked in red in the Results and Discussion Section (page 7) of the revised manuscript.
(7) The behavior of the mobility, which first increases and then decreases, is described in the manuscript; it is recommended to add a brief discussion on the underlying physical mechanisms, such as trap formation or increased surface roughness.
Reply: Apologies for the lack of clarity in our manuscript, which may have led to a delay in understanding this issue. We have explained this behavior on page 5 of the manuscript, as follows: " The thickness dependence of mobility can be attributed to that the thinner channel on the substrate is more susceptible to potential variations due to the formation of traps at the interface, which has been observed in other layered 2D materials. The mobility increases monotonically with the thickness of BP ranging from 1 nm to 8 nm, attributed to the enhanced charge screening. However, the mobility continuously decreases when the thickness increases from ~10 nm to ~20 nm, derived from the rise of interlayer resistance for BP. " We will revise and highlight the section in red to make it more intuitive and easier to understand.
- Include information in the manuscript on why those specific exposure times (e.g., 12, 24, 36, 84 h) were chosen for dry oxygen treatment, and whether the process is expected to be continuous or reach a saturation point.
Reply: We sincerely thank the reviewer for their thoughtful suggestion regarding the specific exposure times used in the dry oxygen treatment. In fact, we conducted multiple tests and selected key time points, such as 12, 24, 36, and 84 hours, based on the observed changes in the BP FET characteristics, particularly at the turning points where significant device performance changes occurred. For instance, after 36 hours of exposure, the hole mobility sharply increased from ~235 cm²/V·s to ~851 cm²/V·s. After this, up to 60 hours, it gradually decreased, which can be considered as reaching saturation. After 96 hours of dry exposure, as shown in Fig. R1, the BP FET no longer exhibited distinct transport characteristics, indicating device failure and loss of FET functionality. Furthermore, after 108 hours of exposure, the on-state current dropped to only 0.1 nA, primarily due to leakage currents from the gate dielectric layer, indicating that the conductive channel had been completely oxidized. From these observations, we conclude that the dry oxidation process does not indefinitely improve performance. Rather, it reaches a saturation point beyond which further oxidation results in degradation of the device’s electrical performance. The data clearly shows that the performance initially improves with oxidation time, but after a certain threshold , it begins to decline.
Fig. R1 Transfer characteristics of BP FET in dry oxidation process after 96 h and 108 h.
- How many experimental replicates were conducted for each exposure condition? Are the presented values averages or individual measurements?
Reply: We thank the reviewer for raising this important question regarding the number of experimental replicates and the presentation of data in the manuscript. For the BP oxidation experiments, multiple replicates were indeed performed under each exposure condition. However, for clarity and to highlight the systematic trend observed during the oxidation process, we chose to present data from a representative experiment. In all of our experiments, we consistently observed the same trend of performance improvement followed by degradation as oxidation time increased. The values presented in the manuscript are individual measurements from representative experiment, rather than averages of multiple replicates. This approach was chosen to provide a clear and direct view of the observed trend, as all replicates exhibited similar behavior.
- Was the relationship between thickness reduction and exposure time quantified directly using AFM?
Reply: Thanks for the reviewer’s comment. In this study, we did not directly use AFM to quantify the thickness reduction over time for several important reasons. First, AFM measurements were performed in ambient conditions rather than in vacuum, which could introduce interference from the surrounding environment, potentially affecting the oxidation process during the testing. The AFM testing process itself could influence the BP oxidation and introduce artifacts into the results.
Additionally, and AFM was only used to measure the initial thickness of the BP device. AFM is not suitable for tracking the layer-by-layer thinning process over time because, after oxidation, the dense BP surface forms amorphous PxOy, which makes it a thickening process over time. In addition, The presence of both BP and PxOy at the surface makes it difficult to discern the exact morphology and thickness of BP during the measurement.
(11) Add to the manuscript whether permanent structural changes (e.g., defects, cracks) were evaluated after 84 h of exposure using SEM or TEM.
Reply: We appreciate the reviewer’s comment regarding the evaluation of permanent structural changes (e.g., defects or cracks) after 84 hours of BP exposure.
Regarding the observation of permanent structural changes, while we did not perform direct SEM or TEM analyses after 84 hours of exposure in this study, we did rely on complementary techniques such as optical and electrical characterizations to track the degradation of BP under oxygen exposure. Specifically, we observed significant performance degradation of BP FET devices, including a sharp decline in the on-state current after extended exposure (see Fig R1). The optical analysis also indicated complete oxidation of the conductive channel in the BP FET after 108 hours of exposure, corroborating these findings.
Additionally, previous studies have provided insights into the structural degradation of BP upon exposure to oxygen. For example, Gómez-Pérez et al. described the oxidation of BP and its impact on the electronic properties and structural changes at various stages, including the formation of different oxide layers, which likely induced permanent changes (ACS omega, 2018, 3(10): 12482-12488). Also, Yao et al. demonstrated edge reconstruction and evolution in BP using in situ high-resolution TEM, showing that such structural changes, including defects and reconstructions, could significantly affect BP's properties (Nanoscale, 2021, 13(7): 4133-4139). Given the experimental setup, we did not use TEM to monitor these structural changes directly after the 84-hour exposure, primarily because of the challenge in decoupling the effects of exposure from the inherent impact of the TEM environment itself. However, based on optical and electrical results as well as the findings from the cited literature, it is reasonable to infer that the observed degradation includes structural changes like defect formation and potential cracks.
Fig. R2 Time-sequential HRTEM images of edge reconstruction and structural evolution in ML-Bp. (Nanoscale, 2021, 13(7): 4133-4139)
(12) How does the presence of PxOy affect long-term device performance (e.g., stability, electronic noise)?
Reply: Thanks for the reviewer’s comment.The presence of PxOy on the BP surface plays a significant role in enhancing the stability of BP, especially under ambient conditions. Previous studies have shown that the surface oxide layer can indeed reduce scattering, thus enhancing stability by acting as a protective barrier against further oxidation (ACS omega, 2018, 3(10): 12482-12488). The formation of PxOy helps mitigate BP degradation by passivating the surface and preventing continuous oxidation of the underlying material. However, the quality of this oxide layer, including its density and uniformity, is critical. If the oxide layer is not dense or robust enough, it may fail to effectively shield the material, leading to possible long-term performance degradation. In some cases, the formation of PxOy has also been linked to improved electronic properties. Surface passivation by oxidation layers can reduce charge scattering, thereby improving the mobility and reducing electronic noise, which is beneficial for the stability and performance of BP-based devices. However, it is essential to note that while PxOy can be beneficial for BP's stability, the effectiveness of the passivation layer depends on the type and density of the oxide layer. A denser, more uniform oxide layer tends to provide better long-term stability, whereas a less robust layer could increase electronic noise and lead to instability over time. This is consistent with our observations, where BP devices covered with a native oxide layer showed stable performance compared to those with plasma-induced oxidation, which led to more degradation.
(13) What is the influence of the type of metal contact (Ti/Au) on the observed behavior? Could there be unaccounted Schottky effects?
Reply: Thank you for the reviewer’s comment. In our study, Ti/Au contacts were used to enhance the adhesion between the metal and BP surface. Titanium (Ti) provides a strong interface with BP, improving the overall stability and reducing the risk of electrode detachment, which is crucial for maintaining consistent device performance. The Au layer was chosen for its excellent electrical properties, providing good conductivity with minimal impact on interface resistance.
However, we acknowledge that the use of Ti/Au contacts could potentially introduce Schottky barrier effects due to the work function mismatch between the metal and BP. These effects are typically related to the formation of a Schottky barrier at the metal-semiconductor interface, which can influence carrier injection and potentially contribute to the observed current-voltage characteristics.In other literature, both Cr/Au (Nanomaterials 13.18 (2023): 2607, Advanced Electronic Materials 9.3 (2023): 2201126) and Ti/Au (Chem. Mater. 2016, 28(22), 8330-8339, Materials, 2022, 15 (2): 615-615, ACS Appl. Mater. Interfaces 2018, 10(22), 19069-19075) ontacts have been used for BP devices, but in our study, we specifically used Ti/Au. Although we did not specifically consider Schottky effects in this study, we believe that the oxidation-induced effects play a more significant role in the device behavior than Schottky effects. However, it is possible that Schottky barriers contribute to the overall performance, and this could be explored in future studies.
(14) Was it verified whether the thinning is uniform or localized? Were AFM or Raman maps taken in different sample areas?
Reply: Thanks for the reviewer’s comment. In air, the oxidation of BP is typically localized and gradually progresses until it is fully oxidized. However, our oxidation process occurs more slowly and gently compared to oxidation in air, making it relatively uniform. This oxidation process is similar to the oxidation of WSe2 in ozone (Nano letters, 2015, 15 (3): 2067-73) , as shown in Fig. R3. Our AFM measurements, conducted only at the start of the experiment, assessed the initial BP thickness. However, AFM is not suitable for tracking the layer-by-layer thinning over time, as oxidation forms a dense amorphous PxOy layer on the BP surface, causing thickening. Additionally, the presence of both BP and PxOy makes it difficult to accurately determine the morphology and thickness during measurements. To monitor the oxidation and thinning process over time, we utilized Raman spectroscopy. As shown in Fig. R4, the Raman spectra were measured at different times, providing information on how the material evolves as it thins. The Raman analysis reflects the gradual thinning of BP, with characteristic shifts observed as oxidation progresses. Thus, although oxidation may occur locally, we found that the thinning process for few-layer BP was relatively uniform, and Raman spectroscopy provided a reliable method to track these changes over time.
Fig. R3 The oxidation process of WSe₂ in ozone. With further O3 exposure, the oxide regions coalesce and oxidation terminates leaving a uniform thickness oxide flm on top of unoxidized WSe2 (Nano letters, 2015, 15 (3): 2067-73).
Fig. R4 A local magnification diagram of peaks.
We hope that our response and revision have addressed the editors’ and reviewers’ comments, so as to make our manuscript qualified for publication in Nanomaterials.
Sincerely yours,
Prof. Xiaoming Zheng (on behalf of all authors)
School of Physics and Electronic Science, Hunan University of Science and Technology, Xiangtan, 411201, Hunan , China
Tel: 15874954147
E-Mail: 1080098@hnust.edu.cn

Round 2
Reviewer 1 Report
Comments and Suggestions for Authors
The authors addressed the reviewer's concerns and the revised manuscript can be accepted for publication.
Reviewer 2 Report
Comments and Suggestions for Authors
The author made all changes